# Susceptibility rhythm to bacterial endotoxin in myeloid clock-knockout mice

Veronika Lang[1], Sebastian Ferencik[1], Bharath Ananthasubramaniam[1,2], Achim Kramer[1], Bert Maier[1]*

[1]Laboratory of Chronobiology, Charité Universitätsmedizin Berlin, Berlin, Germany; [2]Institute for Theoretical Biology, Humboldt-Universität zu Berlin, Berlin, Germany

**Abstract** Local circadian clocks are active in most cells of our body. However, their impact on circadian physiology is still under debate. Mortality by endotoxic (LPS) shock is highly time-of-day dependent and local circadian immune function such as the cytokine burst after LPS challenge has been assumed to be causal for the large differences in survival. Here, we investigate the roles of light and myeloid clocks on mortality by endotoxic shock. Strikingly, mice in constant darkness (DD) show a threefold increased susceptibility to LPS as compared to mice in light-dark conditions. Mortality by endotoxic shock as a function of circadian time is independent of light-dark cycles as well as myeloid CLOCK or BMAL1 as demonstrated in conditional knockout mice. Unexpectedly, despite the lack of a myeloid clock these mice still show rhythmic patterns of pro- and anti-inflammatory cytokines such as TNF$\alpha$, MCP-1, IL-18, and IL-10 in peripheral blood as well as time-of-day and site-dependent traffic of myeloid cells. We speculate that systemic time-cues are sufficient to orchestrate innate immune response to LPS by driving immune functions such as cell trafficking and cytokine expression.

*For correspondence:
bert.maier@charite.de

**Competing interests:** The authors declare that no competing interests exist.

## Introduction

Timing of immune-functions is crucial for initiating, establishing, maintaining, and resolving immune-responses. The temporal organization of the immune system also applies for daily recurring tasks and may even help to anticipate times of environmental challenges. In humans, many parameters and functions of the immune system display diurnal patterns (*Scheiermann et al., 2013*), which impact on disease severity and symptoms (*Druzd et al., 2014*). While the concepts of chronobiology are increasingly acknowledged in life-science and medicine (*Callaway and Ledford, 2017*), a deep comprehension of how time-of-day modulates our physiology in health and disease is still lacking.

The fundamental system behind the time-of-day-dependent regulation of an organism, its behavior, physiology and disease is called the circadian clock. In mammals, this clock is organized in a hierarchical manner: a central pacemaker in the brain synchronized to environmental light-dark cycles via the eyes and peripheral clocks receiving and integrating central as well as peripheral (e.g. metabolic) time information. Both central and peripheral clocks are essentially identical in their molecular makeup: Core transcription factors form a negative feedback loop consisting of the activators, CLOCK and BMAL1, and the repressors, PERs and CRYs. Additional feedback loops and regulatory factors amplify, stabilize and fine-tune the cell intrinsic molecular oscillator to achieve an about 24 hr (circadian) periodicity of cell- and tissue-specific clock output functions (*Bass and Lazar, 2016*).

Circadian patterns of various immune-functions have been reported in mice (*Scheiermann et al., 2013*) and other species (*Haus and Smolensky, 1999*; *Wang et al., 2011*; *Lee and Edery, 2008*) including cytokine response to bacterial endotoxin and pathogens, white blood cell traffic (*Méndez-Ferrer et al., 2008*; *Scheiermann et al., 2012*; *Casanova-Acebes et al., 2013*) and natural killer cell activity (*Arjona and Sarkar, 2006*). Cell-intrinsic clocks have been described for many leukocyte subsets of lymphoid as well as myeloid origin, including monocytes/macrophages (*Keller et al., 2009*).

Furthermore, immune-cell intrinsic clocks have been connected to cell-type specific output function such as the TNFα response to LPS in macrophages (*Keller et al., 2009*).

Sepsis is a severe life threatening condition with more than 31 million incidences per year worldwide (*Fleischmann et al., 2016*). Mouse models of sepsis show a strong time-of-day dependency in mortality rate when challenged at different times of the day (*Halberg et al., 1960*; *Feigin et al., 1969*; *Feigin et al., 1972*; *Hrushesky et al., 1994*; *Deng et al., 2018*). Most studies agree about the times of highest (around *Zeitgeber time* [ZT]8 - that is, 8 hr past lights on) or lowest (around ZT20) mortality across different animal models and investigators (*Halberg et al., 1960*; *Hrushesky et al., 1994*; *Feigin et al., 1969*; *Feigin et al., 1972*). The fatal cascade in the pathomechanism of endotoxic shock is initiated by critical doses of LPS recognized by CD14/TLR4-bearing cells, mainly monocytes and macrophages as well as the activation of the complement system. This activation of the innate immune system triggers a massive release of pro-inflammatory cytokines and feeds into a self amplifying loop involving the cytokine release of not only newly recruited leukocytes but also endothelial and other cells (*Hotchkiss et al., 2016*). As a consequence, activation of the coagulation system in endothelial cells, platelets, and plasma as well as increasing endothelial barrier dysfunction promote disseminated intra-vascular coagulation, blood pressure decompensation and final multi-organ dysfunction (*McConnell and Coopersmith, 2016*; *Hotchkiss et al., 2016*).

Here, we investigate the impact of light-dark cycles and local myeloid clocks on time-of-day-dependent survival rates in endotoxic shock using conditional clock-knockout mouse models. We show that peripheral blood cytokine levels as well as mortality triggered by bacterial endotoxin depend on time-of-day despite a functional clock knockout in myeloid cells. Our work thus challenges current models of local regulation of immune responses.

## Results

### Time-of-day-dependent survival in endotoxic shock

Diurnal patterns of LPS-induced mortality (endotoxic shock) have been reported numerous times in different laboratory mouse strains (*Halberg et al., 1960*; *Marpegan et al., 2009*; *Scheiermann et al., 2012*). To address the question, whether time-of-day-dependent susceptibility to LPS is under control of the circadian system rather than being directly or indirectly driven by light, we challenged mice kept either under light-dark conditions (LD 12:12) or in constant darkness (DD) at four different times during the cycles. As expected from previous reports, survival of mice housed in LD was dependent on the time of LPS injection, being highest during the light phase and lowest at night (*Figure 1A*, *Figure 1—figure supplement 1A–C*).

Surprisingly, mice challenged 1 day after transfer in constant darkness using the same dose showed a more than 60% increase in overall mortality compared to mice kept in LD (*Figure 1A*). Furthermore, time-of-day-dependent differences in mortality were much less pronounced under these conditions.

To discriminate, whether constant darkness alters overall susceptibility to LPS leading to a ceiling effect rather than eliminating time-dependent effects, we systematically reduced LPS dosage in DD conditions. By challenging the mice at four times across the day, we determined the half-lethal dose of LPS in constant darkness (*Figure 1B*). As suspected, mice in DD showed a threefold increased susceptibility to LPS as similar doses of LPS result in largely different mortality rates in LD vs. DD (e.g. LD, 15 mg/kg LPS, 9% mortality vs DD, 13 mg/kg LPS, 67% mortality). For circadian rhythm analysis, we extended the number of time points at approximately half-lethal doses both in LD and DD groups, respectively. This revealed diurnal/circadian patterns in mortality rate in LD (p-value=0.06) as well as in DD conditions (p-value=0.001) (*Figure 1C*) demonstrating that the circadian system controls susceptibility to LPS. In addition, this susceptibility is overall increased in constant darkness (*Figure 1A,B* and *Figure 1—figure supplement 1D*).

### Circadian cytokine response upon LPS challenge in mice

We and others have recently found that cells in the immune system harbor self-sustained circadian oscillators, which shape immune functions in a circadian manner (*Keller et al., 2009*; *Hayashi et al., 2007*). Pro-inflammatory responses in murine ex vivo macrophage culture (*Keller et al., 2009*; *Bellet et al., 2013*) are controlled by cell-intrinsic clocks and are most prominent during the day and

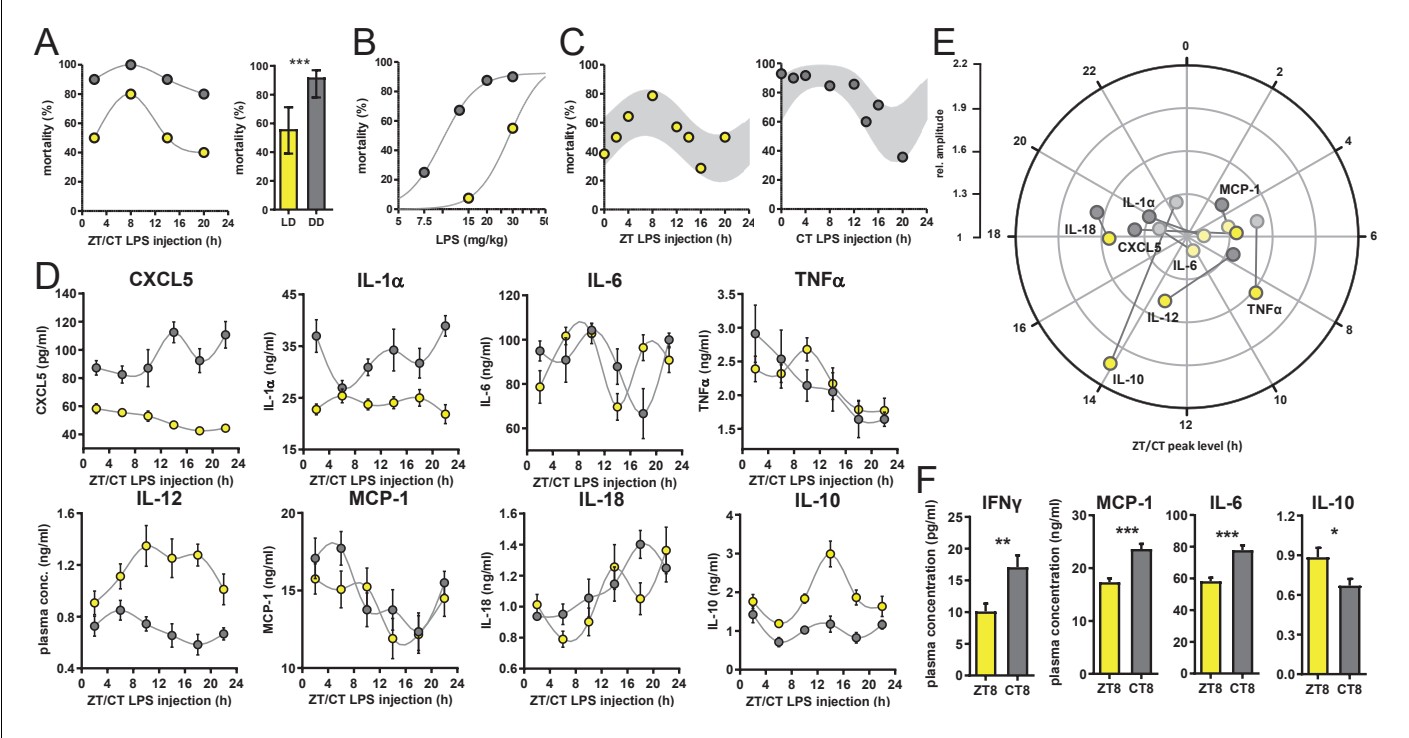

**Figure 1.** Time-of-day-dependent mortality in LPS-treated mice is controlled by the circadian system and light conditions. (**A**) LPS (30 mg/kg, *i.p.*) induced mortality in C57Bl/6 mice (n=10 per time point) kept either in LD 12:12 (yellow) or in DD (gray). Left graph: single time points plottet in either Zeitgeber Time [ZT] or Circadian Time [CT] depending on respective light conditions; right graph: mean mortality of LD or DD light condition. Error bars represent 95% confidence intervals (n=40 per group, *** p=0.0005). (**B**) LPS dose-mortality curves of mice challenged at four time points in LD versus DD (overall n=40 mice per dosage). Gray lines were calculated by fitting an allosteric model to each group. (**C**) Mice (n=10–14 per time point) were challenged with half-lethal doses of LPS (30 mg/kg, *i.p.*), for mice kept in LD (left panel) or 13 mg/kg, *i.p.*, for mice kept in DD (right panel). Mortality was assessed 60 hr after LPS injection. To perform statistical analyses, mortality rates were transformed to probability of death in order to compute sine fit using logistic regression and F-test (LD, p=0.06; DD, p=0.001; gray-shaded areas indicate 95% confidence intervals). (**D**) and (**E**) Time-of-day-dependent cytokine profiles in peripheral blood of C57Bl/6 mice (n=14 per time point) challenged with half-lethal doses of LPS (30 mg/kg, *i.p.*, for mice kept in LD (yellow) or 13 mg/kg, *i.p.*, for mice kept in DD (gray)) and sacrificed 2 hr later. (**E**) Relative amplitudes and phases of cytokines shown in (**D**). Light-colored circles represent non-significant circadian rhythms (p-value>0.05 ) as determined by non-linear least square fit and consecutive F-test (see also Materials and methods section). (**F**) Cytokine levels in peripheral blood from mice (n=10 per time point) challenged with LPS (13 mg/kg, *i.p.*) at either ZT8 (LD) or CT8 (DD) conditions (T-test, *** p<0.001, ** p<0.01, * p<0.05).

The online version of this article includes the following figure supplement(s) for figure 1:

**Figure supplement 1.** Symptomatic characterization of survivors and non-survivors in endotoxic shock mice.

lowest during the activity phase. Furthermore, pro-inflammatory cytokines such as TNFα, IL-1α/β, IL-6, IL-18 as well as MCP-1 have been linked to the pathomechanism of endotoxic shock (*Xing et al., 1998*; *Andrews et al., 2011*; *Gomes et al., 2006*; *Li et al., 1995*; *Marino et al., 1997*).

Thus, we hypothesized that the cytokine response in LPS-challenged mice has a time-of-day dependent profile, which might govern mortality rate in the endotoxic shock model. Indeed, plasma of mice collected 2 hr after administration of half-lethal doses of LPS (either in LD [30 mg/kg] or DD [13 mg/kg] conditions at various times during the day), exhibited an up to twofold time-of-day difference in absolute cytokine concentrations. In animals kept in LD, TNFα showed highest levels around ZT8, IL-18 levels peaked at ZT18 and IL-12p40 as well as the anti-inflammatory cytokine IL-10 had their peak-time around ZT14 (*Figure 1D* and *Figure 1—figure supplement 1E*; for amplitude and phase information as determined by sine fit see *Figure 1E* and *Figure 1—figure supplement 1F*). Interestingly, cytokine profiles from DD mice differed substantially from LD profiles: the peak-times of IL-12p40 was phase-advanced by 6 hr, CXCL5 completely reversed its phase, whereas IL-18 remained expressed predominantly in the night. Taken together, cytokine profiles of LPS-challenged

mice parallel endotoxic shock-induced mortality patterns, although most circadian cytokines show variable phase relations between free-running and entrained conditions (*Figure 1E*).

Next, we investigated, whether the increased overall mortality in DD was correlated with an increased pro-inflammatory cytokine response. We thus injected mice at either ZT8 (mice kept in LD) or circadian time (CT) 8 (mice kept in DD) with the same dose of LPS (13 mg/kg) and took blood samples 2 hr later. IFNγ, MCP-1, IL-6 and the anti-inflammatory cytokine IL-10 showed significantly altered levels between mice kept in LD or DD (*Figure 1F*), suggesting that in DD conditions the sensitivity to endotoxin is increased leading to enhanced pro-inflammatory cytokine secretion and subsequently increased mortality.

## Dispensable role of myeloid clocks in circadian endotoxin reactivity

Local clocks are thought to play important roles in mediating circadian modulation of cell- and tissue-specific functions (*Stratmann and Schibler, 2006*). In fact, depletion of local immune clocks has been shown to disrupt circadian patterns of tissue function (*Nguyen et al., 2013*; *Curtis et al., 2015*; *Gibbs et al., 2012*; *Gibbs et al., 2014*; *Druzd et al., 2017*). To test whether local clocks in cells of the innate immune system are responsible for circadian time dependency in the response to bacterial endotoxin, we challenged myeloid lineage *Bmal1*-knockout mice (*Lyz2 cre/cre* x *Bmal1 flox/flox*, hereafter called myBmal-KO) kept in DD with half-lethal doses of LPS at various times across the circadian cycle. These mice lack physiological levels of Bmal1 mRNA and protein in cells of myeloid origin (*Figure 2—figure supplement 1A–C*) and have been characterized in more detail elsewhere (*Gibbs et al., 2012*). To our surprise, mortality of these mice was still dependent on circadian time of LPS administration (*Figure 2A*) indicating that a functional circadian clock in myeloid cells is not required for time-of-day-dependent LPS-sensitivity. However, the overall susceptibility to LPS decreased twofold compared to wild-type mice (*Figure 2B,C* and *Figure 2—figure supplement 1E*) suggesting that BMAL1 levels in myeloid cells directly or indirectly modulate susceptibility toward LPS. The latter effect could only in part be attributed to genetic background (note decreased susceptibility to LPS in *Lyz2 cre/cre*) (hereafter called LysM-Cre) control mice (*Figure 2C* and *Figure 2—figure supplement 1D*) together arguing for a tonic rather than temporal role of myeloid BMAL1 in regulating LPS sensitivity. Furthermore, switching light conditions from LD to DD in myBmal-KO increased sensitivity to LPS similar to wild-type mice (*Figure 2D*).

Despite its essential role for circadian clock function, BMAL1 has been linked with a number of other non-rhythmic processes such as adipogenesis (*Shimba et al., 2005*), sleep regulation (*Ehlen et al., 2017*), and cartilage homeostasis (*Dudek et al., 2016*). Thus, we asked whether the decreased susceptibility to LPS observed in myBmal-KO mice was due to non-temporal functions of BMAL1 rather than to the disruption of local myeloid clocks. CLOCK, like its heterodimeric binding partner BMAL1, has also been shown to be an indispensable factor for peripheral clock function (*DeBruyne et al., 2007*). If non-temporal outputs of myeloid clocks rather than gene-specific functions of myeloid BMAL1 controls the susceptibility to LPS, a depletion of CLOCK should copy the myBmal-KO phenotype. We therefore generated conditional, myeloid lineage-specific Clock-KO mice (hereafter called myClock-KO). As in myBmal1-KO mice, the expression of Cre recombinase is driven by a myeloid-specific promoter (LysM) consequently leading to excision of LoxP flanked exon 5 and 6 of the clock gene (*Debruyne et al., 2006*). myClock-KO mice showed normal locomotor activity levels and circadian rhythm periods as compared to control mice (*Figure 3—figure supplement 1A–C*). As expected, the expression of *Clock* mRNA and protein was substantially reduced in peritoneal cavity cells but was normal in liver (*Figure 3A,B*).

To investigate, whether clock gene rhythms were truly abolished in myClock-KO mice, we harvested peritoneal macrophages from myClock-KO or control mice (LysM-Cre) kept in DD in regular 4 hr intervals over the course of 24 hr. Rhythmicity of *Bmal1*, *Cry1*, *Cry2*, *Dbp*, *Npas2*, and *Nr1d1* mRNA levels was essentially eliminated, while a low amplitude rhythmicity was detected for *Per1* and *Per2* mRNA (*Figure 3C,D* and *Figure 3—figure supplement 1D,E*). Moreover, circadian oscillations were disrupted in peritoneal cavity cells (mainly macrophages and B-cells) from myClock-KO mice, but not in tissue explants from SCN or lung (*Figure 3E*).

Given the disruption of rhythmicity in myClock-KO myeloid cells, we asked whether CLOCK in myeloid cells is required for time-of-day-dependent mortality in endotoxic shock. To this end, we challenged myClock-KO mice kept in DD at various times during the circadian cycle with half-lethal doses of LPS. Again, mortality in these mice was significantly time-of-day dependent (*Figure 3F*). As

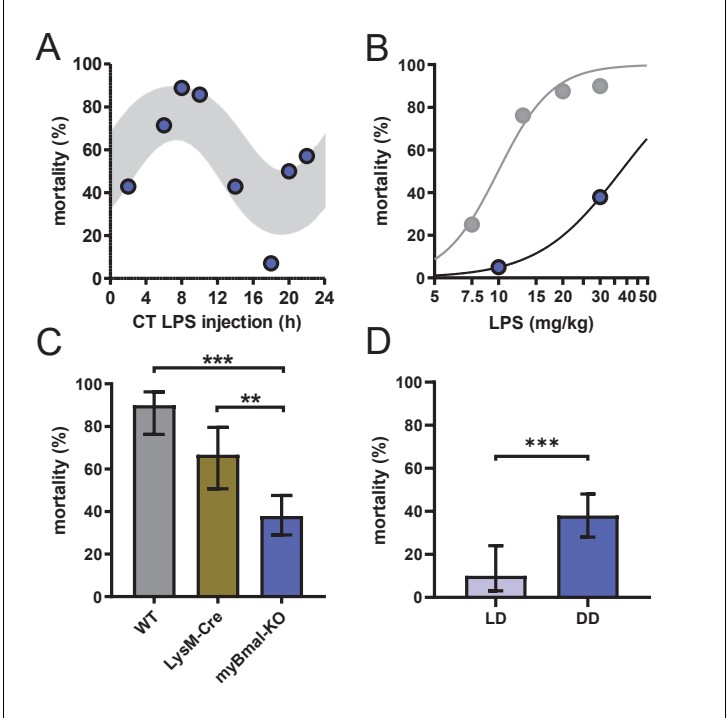

**Figure 2.** myBmal-KO mice show decreased and circadian time-dependent susceptibility to LPS. (A) Circadian mortality in myBmal-KO mice. Mice kept in DD (n=10–14 per group) were challenged with half-lethal doses of LPS (30 mg/kg, *i.p.*) at indicated time points. Mortality was assessed 60 hr after LPS injection. Statistics were performed as in *Figure 1C*, (p=0.0009; gray shaded area indicates 95% confidence interval). (B) LPS dose-mortality curves of mice challenged at 4–8 time points across 24 hr in constant dark conditions. About threefold decrease of susceptibility to LPS in myBmal-KO mice (blue circles) as compared to wild-type mice (gray circles – re-plotted from *Figure 1B* kept in DD). Gray lines were calculated by fitting an allosteric model to each group. (C) Reduced mean mortality in myBmal-KO mice (n=103) compared to control strains LysM-Cre (n=39) or C57Bl/6 wild-type (WT, n=40). All mice were kept in DD and challenged with 30 mg/kg LPS *i.p.* Mean values and 95% confidence intervals from WT and myBmal-KO mice were calculated from experiments shown in *Figure 1C* (WT) and (A) (myBmal-KO)(** p=0.0021, *** p¡0.0001). (D) Constant dark conditions render mice more susceptible to LPS (30 mg/kg LPS) independent of Bmal1 in myeloid lineage cells (n=40 (LD) and n=103 (DD)). ( ** p<0.01, *** p<0.001). The online version of this article includes the following source data and figure supplement(s) for figure 2:

**Figure supplement 1.** Molecular characterization of myBmal-KO mice.

**Figure supplement 1—source data 1.** Raw imaging data files from conditional knockout mouse genotyping.

in myBmal1-KO mice, myClock-KO mice showed strongly reduced susceptibility to LPS compared to wild-type and to a lower degree to LysM-Cre control mice (*Figure 3G* and *Figure 3—figure supplement 1F*). Taken together, our data unequivocally show that myeloid clockwork are dispensable for the time-of-day dependency in endotoxic shock. In addition, decreased overall susceptibility suggest a non-temporal, sensitizing role of myeloid CLOCK/BMAL1 in the regulation of endotoxic shock.

## Circadian cytokine response in myClock-KO mice

Our initial hypothesis was built on the assumption that a time-of-day-dependent cytokine response determines the outcome in endotoxic shock. Previous results from us and others (*Keller et al., 2009*; *Hayashi et al., 2007*; *Gibbs et al., 2012*) suggested that local myeloid clocks govern the timing of the pro-inflammatory cytokine response. However, circadian mortality profiles in LPS-challenged myeloid clock-knockout mice (*Figure 2A* and *Figure 3F*) led us to question this model: The circadian cytokine response in plasma is either independent of a myeloid clock or the circadian mortality by endotoxic shock does not require a circadian cytokine response.

To test these mutually not exclusive possibilities, we challenged myClock-KO mice kept in DD with half-lethal doses of LPS in regular 4 hr intervals over the course of one day. Unexpectedly,

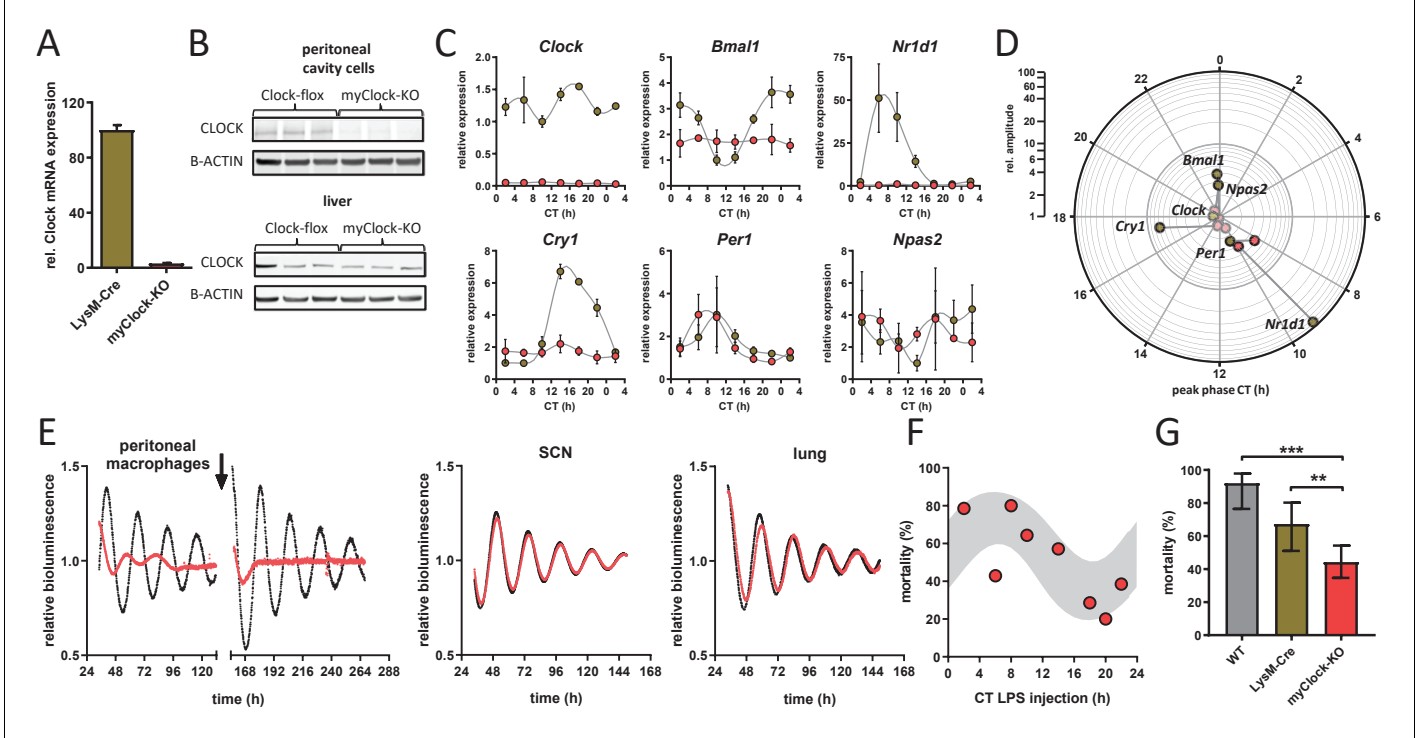

**Figure 3.** Conditional myClock-KO mice show circadian pattern in mortality by endotoxic shock. (**A**) and (**B**) Reduced levels of *Clock* mRNA and protein in myeloid lineage cells of myClock-KO mice. (**A**) Mean values of normalized mRNA expression values of time-series shown in (**C**) (n=21 mice per condition, p<0.0001, t-test). (**B**) Protein levels by immunoblot in peritoneal cavity cells and liver of myClock-KO and *Clock*^flox/flox^ control mice (n=3). (**C**) Relative mRNA levels of selected clock genes in peritoneal macrophages from LysM-Cre (brown circles) or myClock-KO (red circles) mice, both kept in DD, at indicated circadian times. Phase and amplitude information are depicted in (**D**) as analyzed by Chronolyse. Non-significant circadian expression (p>0.05) are depicted in light red (myClock-KO) or light brown (LysM-Cre control). (**E**) Representative bioluminescence recordings of peritoneal macrophages, SCN or lung tissue from myClock-KO or wild-type mice crossed with PER2:Luc reporter mice (color coding as before). Black arrow indicates time of re-synchronization by dexamethasone treatment (detrended data). (**F**) Circadian pattern in endotoxic shock mortality despite deficiency of CLOCK in myeloid lineage cells. Mice (DD, n=10–14 per time point) were challenged with half-lethal doses of LPS (30 mg/kg, *i.p.*) at indicated time points. Mortality was assessed 60 hr after LPS injection. Statistic were performed as in *Figure 1C* (p=0.005, gray shaded area indicates 95% confidence interval). (**G**) Reduced mean mortality (at 30 mg/kg LPS) in mice deficient of myeloid CLOCK (n=103) compared to control strains LysM-Cre (n=39) or C57Bl/6 (wild-type, n=40). First two bars where re-plotted from *Figure 2C*. Error bars represent 95% confidence intervals (* p=0.0192, *** p = 0.0001).

The online version of this article includes the following source data and figure supplement(s) for figure 3:

**Source data 1.** Raw imaging data files from western blots.

**Figure supplement 1.** Phenotypic and molecular characterization of myClock-KO mice.

cytokine levels in plasma, collected 2 hr after LPS administration still exhibited circadian patterns for TNFα, IL-18, IL-10 (*Figure 4A*) (p-values=0.046, 0.001 and 0.009, respectively), very similar to those observed in wild-type animals (*Figure 4B*). Other cytokines remained below statistical significance threshold for circadian rhythmicity tests (IL-1α) or displayed large trends (IL-6) within this period of time (see also *Figure 4—figure supplement 1A,B*). These data suggest that an LPS-induced circadian cytokine response does not depend on a functional circadian clock in cells of myeloid origin. Thus, circadian cytokine expression might still be responsible for time-of-day-dependent mortality in endotoxic shock.

To identify those cytokines, whose levels best explain mortality upon LPS challenge, we correlated cytokine levels and mortality rate for all conditions – independent of time-of-day or mouse strain – in a linear correlation analysis. Levels of CCL7, MCP-1 and TNFα showed strong positive correlation with mortality (p-values=0.0003, 0.0065, 0.0319, respectively). Others, such as IL-10 and IL-18 correlated negatively (p-values=0.0054 and 0.0004, respectively) (*Figure 4C* and *Figure 4—figure supplement 1C*). Interestingly, while TNFα and IL-10 are well-known factors in the

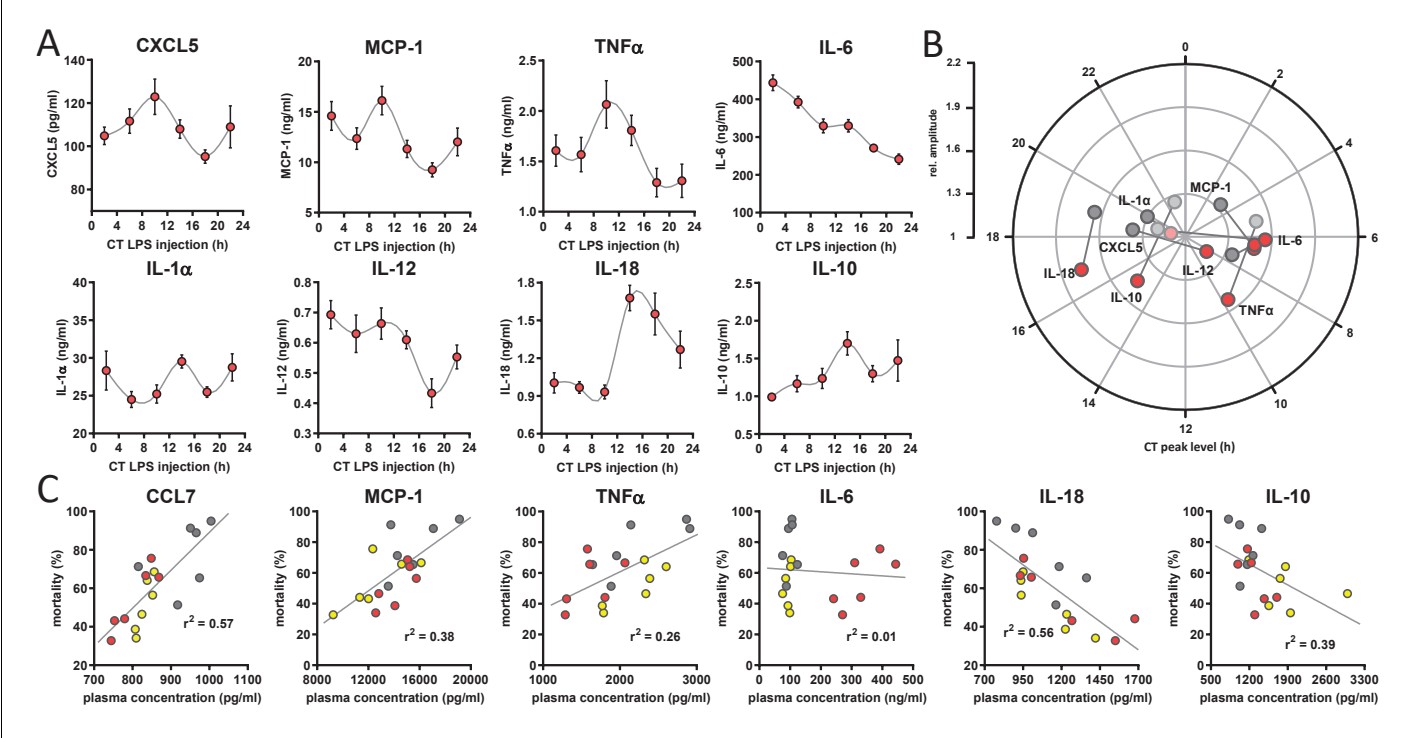

**Figure 4.** Circadian time-dependent cytokine levels in plasma of myClock-KO mice. (**A**) Plasma cytokine levels in myClock-KO mice kept in constant darkness, 2 hr after *i.p.* injection of 30 mg/kg LPS. Data represent mean values ± SEM (n=14 per time point). (**B**) Polar plot showing amplitude and phase distribution of pro- and anti-inflammatory cytokines from A, red circles and wild-type DD (*Figure 1E*). Light circles indicate non-significant circadian abundance (p-values>0.05, non-linear least square fit statistics by Chronolyse). (**C**) Overall correlation of cytokine levels with mortality independent of time-of-day of LPS injection and mouse model. Colors indicate data source (wild-type, LD - yellow; wild-type, DD - gray; myClock-KO, DD - red; linear regression - gray line; statistics: spearman correlation).

The online version of this article includes the following figure supplement(s) for figure 4:

**Figure supplement 1.** Circadian time-dependent cytokine levels in plasma of myClock-KO mice.

pathomechanism of endotoxic shock, CCL7, MCP-1 and protective effects of IL-18 have not been reported in this context.

Together, our data suggest that local circadian clocks in myeloid lineage cells are dispensable for time-of-day-dependent plasma cytokine levels upon LPS challenge. However, where does time-of-day dependency in endotoxic shock originate instead?

## Persistent circadian traffic in myeloid clock-knockout mice

Circadian patterns in immune cell trafficking and distribution have been recently reported and linked to disease models and immune functions (*Keller et al., 2009*; *Nguyen et al., 2013*; *Druzd et al., 2017*). Similarly, homing and release/egress of hematopoetic stem cells (HSPCs), granulocytes, and lymphocytes to bone marrow and lymph nodes, respectively, have been shown to vary in a time-of-day-dependent manner requiring the integrity of an immune-cell intrinsic circadian clock (*Druzd et al., 2017*; *Méndez-Ferrer et al., 2008*; *Scheiermann et al., 2012*; *Casanova-Acebes et al., 2013*). Thus, we asked, whether circadian traffic of myeloid cells can be associated with mortality rhythms in our endotoxic shock model.

To test this, we measured the number of immune cells in various immunological compartments at two distinct circadian time points representing peak (CT8) and trough (CT20) of circadian mortality rate upon LPS challenge. Cells from wild-type, genotype control and myeloid clock-knockout mice (n=5 per time point, all mice kept in DD) were collected from a broad spectrum of immune system compartments (blood, peritoneal cavity, bone marrow, spleen, inguinal lymph nodes, thymus) (*Figure 5A*). As expected from previous studies (*Druzd et al., 2017*; *Keller et al., 2009*;

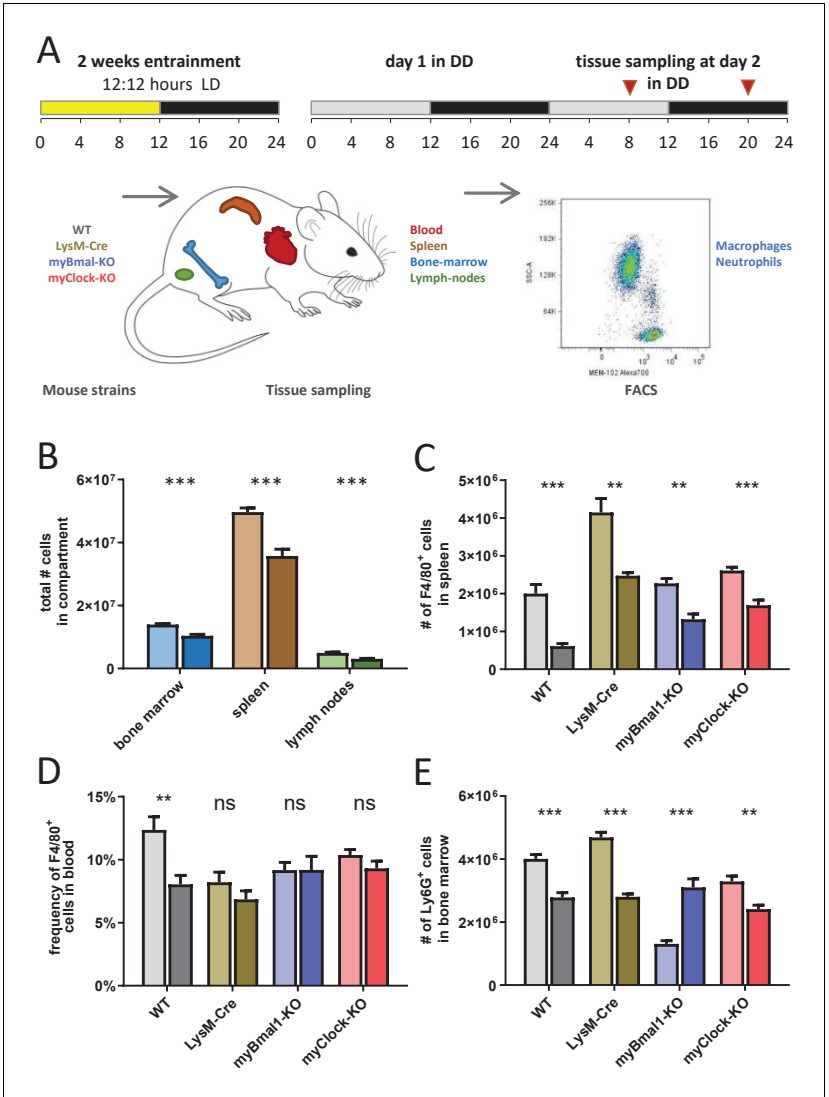

**Figure 5.** Time-of-day-dependent traffic of myeloid cells despite depletion of myeloid CLOCK or BMAL1. (**A**) Experimental scheme to investigate time-of-day-dependent immune cell traffic in various compartments and genetic mouse models. (**B**) Total cell counts of femoral bone marrow (blue), spleen (brown), or inguinal lymph nodes (green) at CT8 (light colors) or CT20 (dark colors). (**C–E**) Cell number or frequency from wild-type and various conditional circadian clock knockout mice kept in DD at two different circadian time points (wild-type - gray, LysM-Cre - brown, myBmal-KO - blue, myClock-KO - red, CT8 - light, CT20 - dark). (**C**) Total number of F4/80$^+$ macrophages in spleen. (**D**) Relative number of F4/80$^+$ macrophages in blood. Asterisks indicate level of significance as determined by t-test. (**E**) Total number of Ly6G$^+$ neutrophils in bone marrow (significance levels: ns $p>0.05$, * $p<0.05$, ** $p<0.01$, *** $p<0.001$).

The online version of this article includes the following figure supplement(s) for figure 5:

**Figure supplement 1.** FACS gating strategy for spleen-derived cells.
**Figure supplement 2.** FACS gating strategy for bone-marrow-derived cells.
**Figure supplement 3.** FACS gating strategy for lymph-node-derived cells.
**Figure supplement 4.** FACS gating strategy for thymus-derived cells.
**Figure supplement 5.** FACS gating strategy for blood-derived cells.
**Figure supplement 6.** FACS gating strategy for peritoneal-cavity-derived cells.
**Figure supplement 7.** Cell numbers of myeloid panel from bone marrow, spleen, and lymph nodes.
**Figure supplement 8.** Cell numbers of lymphoid panel from bone marrow, spleen, and lymph nodes.
**Figure supplement 9.** Cell frequencies of myeloid panel from blood, peritoneal cavity, and thymus.
**Figure supplement 10.** Cell frequencies of lymphoid panel from blood, peritoneal cavity, and thymus.

*Scheiermann et al., 2012*) the number of cells were significantly time-of-day dependent in bone marrow, spleen, and inguinal lymph nodes of wild-type mice (*Figure 5B*, *Figure 5—figure supplements 7–10*).

If circadian traffic of myeloid cells and therefore lymphoid organ composition would be a main factor in regulating sensitivity to bacterial endotoxin, similar patterns in myeloid clock-knockout mice should be observed. However, results obtained from myBmal-KO and myClock-KO mice did not support this hypothesis. While in spleen, F4/80+ macrophages of both myeloid clock knockout strains were still found at higher numbers at CT8 compared to CT20 (*Figure 5C*), significant time-of-day differences of monocyte/macrophage numbers in wild-type blood diminished in myeloid clock knockouts (*Figure 5D*). Different patterns of time-of-day dependency were also observed in bone marrow (*Figure 5E*), together suggesting a rather complex than mono-causal relation between myeloid clocks, circadian traffic and mortality risk in endotoxic shock (for a comprehensive analysis of this dataset see *Figure 5—figure supplements 7–10*).

## Discussion

In this study, we tested the hypothesis that local myeloid circadian clocks regulate the devastating immune response in endotoxic shock. A number of findings by various labs including our own pointed to such a possibility. First, monocytes/macrophages have been identified as important cellular entities relaying the endotoxin (LPS) triggered signal to the immune system and other organ systems by means of massive secretion of pro-inflammatory cytokines (*Sabroe et al., 2002*; *Novick et al., 2013*). Second, a high amplitude circadian clock in monocytes/macrophages has been shown to control circadian cytokine output in vitro and ex vivo (*Keller et al., 2009*; *Curtis et al., 2015*; *Bellet et al., 2013*). Third, a high amplitude time-of-day-dependent mortality has been demonstrated in endotoxic shock (*Halberg et al., 1960*). Fourth, a number of experiments done at two circadian time points, including a cecal ligation and puncture model (*Deng et al., 2018*), demonstrating loss of time-of-day dependency of mortality in myeloid *Bmal1* knockout mice (*Curtis et al., 2015*) further supported this hypothesis.

On the other hand, some studies suggested other mechanisms, for example Marpegan and colleagues reported that mice challenged with LPS at two different time points in constant darkness did not exhibit differences in mortality rates (*Marpegan et al., 2009*), arguing for a light-driven process in regulating time-of-day dependency in endotoxic shock. However, we suspected that this result may be caused by a ceiling effect induced by constant darkness, since mortality rates at both time points were close to 100 percent. Indeed, when we compared susceptibility of mice challenged with a similar dose of LPS in light-dark versus constant darkness, we observed a marked increase of overall mortality. Similarly, depletion/mutation of *Per2* rendered mice insensitive to experimental time dependency in endotoxic shock (*Liu et al., 2006*). Surprisingly, however, we found that depletion of either CLOCK or BMAL1 in myeloid lineage derived cells both did not abolish time-of-day dependency in mortality to endotoxic shock, which led us to reject our initial hypothesis.

Our data also exclude light as a stimulus driving mortality: First, data from Halberg, Marpegan, Scheiermann as well as our own lab (*Halberg et al., 1960*; *Marpegan et al., 2009*; *Keller et al., 2009*; *Scheiermann et al., 2012*) suggest that mice are more susceptible to endotoxic shock during the light phase compared to dark phase - whereas switching light schedules from light-dark to constant darkness led to an increase in overall mortality. Second, irrespective of the genetic clock-gene depletion tested, our data unequivocally demonstrate circadian mortality rhythms upon LPS challenge even under DD conditions. Strikingly, peripheral blood cytokine levels showed - though altered - circadian time dependency in the myClock-KO strain.

A burst of cytokines, following a lethal dose of LPS is generally thought to be an indispensable factor causing multi-organ dysfunction and leading to death. However, the contribution of single cytokines has been difficult to tease apart due to complex nature of interconnected feedback systems. By adding time-of-day as an independent variable in a number of different mouse models, we were able to correlate individual cytokines' contribution to mortality in the context of the complex response to LPS. While the pro-inflammatory cytokine TNFα was confirmed to act detrimentally, IL-18 surprisingly turned out to likely be protective. This interpretation is not only supported by the negative correlation of IL-18 levels in blood and mortality risk, but also by an anti-phasic oscillation

of IL-18 levels paralleling those of the known protective cytokine IL-10. In addition, a protective role for IL-18 has been previously reported in a colitis model in mice (*Dinarello et al., 2013*).

While circadian regulation of trafficking lymphocytes by cell intrinsic clockworks have been demonstrated to impact the pathophysiology of an autoimmune disease model such as EAE (*Druzd et al., 2017*), our data on distribution patterns of immune cells at two circadian time points draw a more complex picture. In spleen, absolute numbers of macrophages, but not neutrophils and monocytes, are independent of their local clocks. In contrast, patterns of macrophages and neutrophils in peripheral blood and bone marrow (respectively) change upon local myeloid clock depletion. Thus, it appears that myeloid cell traffic is regulated at multiple levels including cell-intrinsic, endothelial and site specific factors. However, our data do not support the hypothesis of myeloid cell distribution being a main factor for time-of-day-dependent mortality risk in endotoxic shock. Hence, what remains as the source of these rhythms?

Following the pathophysiology of endotoxic shock on its path from cause to effect, it is important to note that the bio-availability of bacterial endotoxin injected intraperitoneally depends on multiple factors (i.e. pharmaco-kinetics), many of which themselves might underlie circadian regulation. As a consequence, same doses of LPS administered *i.p.* at different times-of-day might result in highly diverging concentrations at the site of action.

Sensing of LPS by TLR4 is not restricted to myeloid lineage cells (*Hotchkiss et al., 2016*; *Wang et al., 2011*; *Bès-Houtmann et al., 2007*; *Abreu et al., 2002*; *Kato et al., 2004*). Subsequent to *i.p.* injection, peritoneal mesothelial cells, adipocytes and adjacent organs might directly react to LPS and shape the complex response not only of cytokine release, but also in changing their metabolic states and tissue functions. Indeed, a recent study performed in a humanized mouse model identified non-hematopoetic derived cells as major source of many plasma cytokines after cecal ligation and puncture (*Skirecki et al., 2019*). Local circadian clocks of these cells may therefore contribute to the time-of-day dependent cytokine patterns and survival observed in myeloid clock knockout mice.

Along this path, rhythmic feeding behavior driven by the central pacemaker was suggested to alter immune-responses directly or indirectly (*Luna-Moreno et al., 2009*; *Laermans et al., 2014*) and a recent study showed that day-time restricted feeding can reverse time-of-day dependent mortality rates (*Geiger et al., 2021*). Also, nutrition-related factors could drive immune cells to respond differently to stimuli independent or dependent of local clocks (*Nguyen et al., 2013*; *Huang et al., 2007*; *Geiger et al., 2021*).

Finally, the ability of cells and organs to resist all sorts of noxa might as well be regulated in a time-of-day-dependent manner. In this case, rather than same doses of toxin leading to time-dependent cytokine responses (noxa) resulting in diverging rates of multi organ failure, same amount of noxa would cause time-dependent rates of multi organ dysfunction and death. Indeed, work from *Hrushesky et al., 1994* showed that mice challenged with same doses of TNFα at various times throughout the day exhibited time-dependent survival paralleling phenomena in endotoxic shock. However, our data showing fluctuating cytokine levels in myClock-KO mice challenged with LPS in constant darkness (*Figure 4A*) question the mechanism of organ vulnerability as the only source of circadian regulation.

While our work suggests that local myeloid clocks do not account for time-of-day dependent mortality in endotoxic shock it unequivocally argues for a strong enhancing effect of myeloid CLOCK and BMAL1 on overall susceptibility, which adds on the effect of light conditions. However, it is important to note that our data do stay in conflict with findings from other labs which rather reported attenuating effects of BMAL1 on inflammation (*Curtis et al., 2015*; *Nguyen et al., 2013*; *Deng et al., 2018*) but align well with a report on clock mutant mice (*Bellet et al., 2013*). Differences in genetic background of mouse strains, animal-facility-dependent microbiomes (*Sanz and Moya-Pérez, 2014*) or animal care procedures might account for this but remain unsatisfying explanations.

One of the most striking findings of our work is the large increase in susceptibility to endotoxic shock when mice were housed under DD as compared to LD conditions. This relative increase when shifting from LD to DD was observed in wild-type as well as in myBmal-KO mice and appears therefore independent of a functional clock in myeloid cells. Interestingly, Carlson and Chiu reported similar effects in a cecal ligation and puncture model in rats upon transfer to LL (constant light) or DD conditions, where they found decreased survival as compared to rats remaining in LD conditions

(*Carlson and Chiu, 2008*). It is tempting to speculate that rhythmic light conditions, rather than light itself promote survival in endotoxemia. In either case, it will be important to further investigate these phenomena not only in respect to animal housing conditions, which need to be tightly light controlled in immunological, physiological, and behavioral experiments but also for their apparent implications on health-care in intensive care units.

# Materials and methods

## Key resources table

| Reagent type (species) or resource | Designation | Source or reference | Identifiers | Additional information |
|---|---|---|---|---|
| Strain, strain background (*Mus musculus*) | C57Bl/6J | The Jackson Laboratory | MGI:3028467 RRID:MGI:3028467 | |
| Strain, strain background (*Mus musculus*) | B6.129P2-Lyz2tm1(cre)Ifo/JC57Bl/6J | *Clausen et al., 1999* | MGI:1934631 RRID:IMSR_JAX:004781 | |
| Strain, strain background (*Mus musculus*) | B6.129S4(Cg)-Arntltm1Weit/J, C57Bl/6J | *Storch et al., 2007* | MGI: J:124157 | |
| Strain, strain background (*Mus musculus*) | B6.129S4-Clocktm1Rep/J, C57Bl/6J | *Debruyne et al., 2006* | MGI: J:109634 | |
| Strain, strain background (*Mus musculus*) | B6.129S6-Per2tm1Jt/J, C57Bl/6J | *Yoo et al., 2004* | MGI: J:89256 | |
| biological sample (*Escherichia coli* O55:B5) | LPS | Sigma-Aldrich | L2880 | LOTs 102M4017V, 032M4082V, 063M4041V |
| Antibody | CD11b Microbeads (rat monoclonal) | Miltenyi | Cat# 130-049-601 | MACS (1:5) |
| Antibody | α-mCD3 APC-Cy7 (rat monoclonal) | Biolegend | Cat# 100221 RRID:AB_2057374 | FACS (1:200) |
| Antibody | α-mCD4 FITC (rat monoclonal) | Biolegend | Cat# 100405 RRID:AB_312690 | FACS (1:500) |
| Antibody | α-mCD8 PE-Cy7 (rat monoclonal) | Biolegend | Cat# 100721 RRID:AB_312760 | FACS (1:200) |
| Antibody | α-mCD11b PE-Cy7 (rat monoclonal) | Ebioscience | Cat# 25-0112 RRID:AB_469587 | FACS (1:200) |
| Antibody | α-mCD11c APC (armenian hamster monoclonal) | Biolegend | Cat# 117309 RRID:AB_313778 | FACS (1:200) |
| Antibody | α-mCD14 PE (rat monoclonal) | Biolegend | Cat# 123309 RRID:AB_940582 | FACS (1:500) |
| Antibody | α-mCD19 Pacific Blue (rat monoclonal) | Biolegend | Cat# 115526 RRID:AB_493341 | FACS (1:200) |
| Antibody | α-mCD45 PerCP-Cy5.5 (rat monoclonal) | Biolegend | Cat# 103131 RRID:AB_893344 | FACS (1:200) |
| Antibody | α-mLy6C Brilliant Violet 421 (rat monoclonal) | Biolegend | Cat# 128033 RRID:AB_2562351 | FACS (1:200) |
| Antibody | α-mLy6G FITC (rat monoclonal) | BD | Cat# 561105 RRID:AB_394207 | FACS (1:1000) |
| Antibody | α-mF4/80 APC (rat monoclonal) | Biolegend | Cat# 123115 RRID:AB_893493 | FACS (1:100) |
| Antibody | α-mNK1.1 PE (mouse monoclonal) | Biolegend | Cat# 108707 RRID:AB_313394 | FACS (1:500) |
| Antibody | VD eFluor 506 | eBioscience | Cat# 65-0866-14 | FACS (1:500) |
| Antibody | VD eFluor 780 | eBioscience | Cat# 65-0865-14 | FACS (1:1000) |

*Continued on next page*

*Continued*

| Reagent type (species) or resource | Designation | Source or reference | Identifiers | Additional information |
|---|---|---|---|---|
| Antibody | Rb α-mBmal1 (rabbit polyclonal) | kind gift from Michael Brunner | | WB (1:500) |
| Antibody | Rb α-mClock (rabbit polyclonal) | Bethyl Laboratories | Cat# A302-618A RRID:AB_10555233 | WB (1:1000) |
| Antibody | Mm α-mβ-Actin (mouse monoclonal) | Sigma-Aldrich | Cat# A-5441 RRID:AB_476744 | WB (1:100000) |
| Antibody | Gt α-mIgG-HRP (goat polyclonal) | SantaCruz Biotechnology | Cat# sc-2005 RRID:AB_631736 | WB (1:1000) |
| Antibody | DK α-rbIgG-HRP (donkey polyclonal) | SantaCruz Biotechnology | Cat# sc-2305 RRID:AB_641180 | WB (1:1000) |
| Commercial assay or kit | Mouse IL-6 ELISA Ready-SET-Go! | Ebioscience | Cat# 88-7064-88 RRID:AB_2574990 | |
| Commercial assay or kit | ProcartaPlex, Mix & Match, mouse 13-plex | Ebioscience | EPX 130-24021-801 | |
| Sequence-based reagent | mCry1 | Qiagen | QT00117012 | |
| Sequence-based reagent | mCry2 | Qiagen | QT00168868 | |
| Sequence-based reagent | mDpb | Qiagen | QT00103089 | |
| Sequence-based reagent | mNr1d1 | Qiagen | QT00164556 | |
| Sequence-based reagent | mNpas2 | Qiagen | QT00108647 | |
| Sequence-based reagent | mPer1 | Qiagen | QT00113337 | |
| Sequence-based reagent | mPer2 | Qiagen | QT00198366 | |
| Sequence-based reagent | mGapdh fw | This paper | qPCR primer | ACGGGAAGCTCACTGGCATGGCCTT |
| Sequence-based reagent | mGapdh rv | This paper | qPCR primer | CATGAGGTCCACCACCCTGTTGCTG |
| Sequence-based reagent | mBmal1 fw | This paper | qPCR primer | GGACACAGACAAAGATGACCC |
| Sequence-based reagent | mBmal1 rv | This paper | qPCR primer | TTTTGTCCCGACGCCTCTTT |
| Sequence-based reagent | mClock fw | This paper | qPCR primer | ATTGGTGGAAGAAGATGACAAGGA |
| Sequence-based reagent | mClock rv | This paper | qPCR primer | TACCAGGAAGCATAGACCCC |
| Sequence-based reagent | Bmal flox com fw | This paper | Genotyping PCR primer | ACTGGAAGTAACTTTATCAAACTG |
| Sequence-based reagent | Bmal flox com rv | This paper | Genotyping PCR primer | CTGACCAACTTGCTAACAATTA |
| Sequence-based reagent | Bmal flox mut fw | This paper | Genotyping PCR primer | CTCCTAACTTGGTTTTTGTCTGT |
| Sequence-based reagent | LysM cre mut rv | This paper | Genotyping PCR primer | CCCAGAAATGCCAGATTACG |
| Sequence-based reagent | LysM cre com fw | This paper | Genotyping PCR primer | CTTGGGCTGCCAGAATTTCTC |
| Sequence-based reagent | LysM cre wt rv | This paper | Genotyping PCR primer | TTACAGTCGGCCAGGCTGAC |
| Sequence-based reagent | Clock flox com fw | This paper | Genotyping PCR primer | CGCTGAGAGCCAAGACAAT |
| Sequence-based reagent | Clock flox com rv | This paper | Genotyping PCR primer | AGCTGGGGTCTATGCTTCCT |
| Sequence-based reagent | Per2 luc com fw | This paper | Genotyping PCR primer | CTGTGTTTACTGCGAGAGT |

*Continued on next page*

*Continued*

| Reagent type (species) or resource | Designation | Source or reference | Identifiers | Additional information |
|---|---|---|---|---|
| Sequence-based reagent | Per2 luc wt rv | This paper | Genotyping PCR primer | GGGTCCATGTGATTAGAAAC |
| Sequence-based reagent | Per2 luc mut rv | This paper | Genotyping PCR primer | TAAAACCGGGAGGTAGATGAGA |
| Software, algorithm | CFX Manager | BioRad | | V2.1 |
| Software, algorithm | Chronolyse | In house generated software | | V2.0 |
| Software, algorithm | Chronostar | In house generated software | https://www.achim-kramer-lab.de/downloads.html | V3.0 |
| Software, algorithm | FlowJo | FlowJo | | V10 |
| Software, algorithm | GraphPad Prism | GraphPad | | V9 |
| Software, algorithm | photoNgraph | In house generated software | | V2.0 |
| Software, algorithm | ClockLab Analysis | Actimetrix | | |

## Animals

All procedures were authorized by and performed in accordance with the guidelines and regulations of the German animal protection law (Deutsches Tierschutzgesetz). Experimental mouse lab was air-conditioned and RT was set to 25°C and 50–60% humidity to exclude time cues and keep environmental conditions constant. Mice were housed in macrolon type II cages supplied with nesting material, food and water ad libitum at a 12 hr:12 hr light/dark (LD) cycle. For endotoxic shock and running wheel experiments mice were individually housed. For all other experiment mice were group-housed. Manipulations during the dark phase of the cycle were performed under infrared light. Male C57Bl/6 mice (Jackson Laboratories strain) mice were purchased from our animal facility (Charite FEM, Berlin, Germany) at 8–10 weeks of age. Homozygous, male *Lyz2*^*Cre/Cre* (LysM-Cre) (*Clausen et al., 1999*), myBmal-KO and myClock-KO were bred and raised in our animal facility (FEM, Berlin Germany) and used at 8–12 weeks. Female *Lyz2*^*Cre/Cre* *Per2:Luc*, either wild-type or homozygous for *Clock*^*flox*, were bred and raised in our animal facility (FEM, Berlin Germany) and used at 14 weeks.

## Generation of myeloid clock knockout mice

*Bmal1*^*flox/flox* (Bmal-flox) (*Storch et al., 2007*) or *Clock*^*flox/flox* (Clock-flox) (*Debruyne et al., 2006*) were bred with LysM-Cre (*Clausen et al., 1999*) to target *Bmal1* or *Clock* for deletion in the myeloid lineage. Offspring were genotyped to confirm the presence of the loxP sites within *Bmal1* or *Clock* and to determine presence of the Cre recombinase. Upon successful recombination the loxP flanked exon 8 of *Bmal1* or in case of *Clock* the floxed exon 5 and 6 were deleted. *Lyz2*^*Cre/Cre* x *Clock*^*flox/flox* were crossed onto a *Per2:Luc* (*Yoo et al., 2004*) background for further characterization in biolumi-nescence reporter assays. All mice have been genotyped before experiments.

## Mouse genotyping

Ear punches were taken to genotype individual mice. Genomic DNA was prepared by incubating ear punch in 200 µl DirectPCR-Tail Lysis reagent including 2 µl proteinase K for 4–16 hr at 55°C on a thermoshaker (550 rpm). Subsequently, sample was heat inactivated for 45 min at 85°C and spun for 5 min @ 15,000 g. Supernatant was used for PCR. Primers as listed in key resource table: LysM-Cre *Goren et al., 2009*, Bmal1-flox (*Storch et al., 2007*), Clock-flox (*Debruyne et al., 2006*), Per2-Luc (*Yoo et al., 2004*).

## Locomotor activity recording

Male homozygous myClock-KO mice (control mice: LysM-Cre and Clock-flox mice), 8–10 weeks of age were individually housed with ad libitum acess to a running wheel in Macrolon typ III cages.

Running wheel activity was recorded for 2 weeks in 12 hr:12 hr light:dark (LD) followed by 2 weeks of constant darkness (DD). Locomotor activity was recorded and evaluated with ClockLab Analysis (ActiMetrix). Circadian Period (tau) and overall activity analysis was performed in DD phase.

### Intraperitoneal LPS injection

*E. coli* LPS (055.B5, Sigma Aldrich) stock solution (10 mg/ml) was diluted to appropriate concentration in sterile PBS and thoroughly vortexed before use. LPS injection was performed *i.p.* using Legato 100 Syringe Pump (KD Scientific). The following settings were applied: Mode: infuse only; syringe: BD, plastic, 5 ml; rate: 2 ml/min. A cannula (26G x 3/8') was attached to Legato 100 by microbore extension line (60 cm, MedEx, Smiths Medical) for the LPS injections.

### Endotoxic shock experiments

Eight to 12 weeks old male mice were entrained to 12 hr:12 hr light-dark cycles for 2 weeks. Dosing of LPS injection was adjusted for individual body weight prior injection. Injection volume did not exceed 10 µl/g body weight. For LD experiments, mice were injected *i.p.* on day 14. For DD experiments, mice were transferred to DD on day 14 and were injected *i.p.* on second day in DD. Animals were kept in the respective lighting conditions until the termination of experiment 60 hr post LPS injection. For all endotoxic shock experiments, human endpoints were applied to determine survival (for definition of human endpoints see respective section).

### Time-of-day-dependent mortality experiments

Endotoxic shock mortality experiments were comprised of two parts for each mouse strain and condition tested: First, lethal dose 50 ($LD_{50}$) of LPS was determined by injecting groups of mice (n=10) at four 6 hr spaced time points. The $LD_{50}$ describes the concentration at which approximately 50% of all animals injected (averaged over all injection time points) survive. This ensures most dynamic range for the detection of potential circadian rhythms. Second, experimentally determined $LD_{50}$ of LPS was used to investigate, whether mortality by endotoxic shock was dependent on time-of-day. To this end, mice (n=14 per group) were injected *i.p.* at six 4 hr spaced time points to increase statistical power for circadian rhythm analysis (see Statistical data analysis section).

### Definition of human endpoints

A scoring system was developed in order to detect irreversibly moribund mice before the occurrence of death by endotoxic shock (*Table 1*). It is based on previous reports by *Liu et al., 2006*; *Shimba et al., 2005* and required further refinements according to our experience. Mice in the endotoxic shock experiments were monitored and scored every 2–4 hr for up to 60 hr post LPS injection (*Figure 1—figure supplement 1A*). In addition, surface body temperature was measured at the sternum every 12 hr and body weight was measured every 24 hr (*Figure 1—figure supplement 1B, C*). Mice with a score of 0–2 were monitored every 4 hr. As of a score of 3, the monitoring frequency was increased to every 2 hr. In addition, softened, moisturized food was provided in each cage as of a score of 3. Weight loss exceeding 20%, three consecutive scores of 4, or one score of 5 served as human endpoints. A mouse was defined as a non-survivor when human endpoints applied and was subsequently sacrificed by cervical dislocation. Mice which did not display any signs of endotoxic shock such as weight or temperature loss and no increasing severeness in score (in total 5 of 632 mice [0.8%]) were excluded from the analysis.

**Table 1.** Scoring system to determine human endpoint in endotoxic shock experiments.

| Score | Behavior, Phenotype |
|---|---|
| 0 | Normal, no behavioral abnormalities |
| 1 | Slightly decreased speed of course of movement |
| 2 | Slightly lethargic |
| 3 | 'Slow motion' movements, ruffled fur, hunched posture |
| 4 | Lethargic, movements are not specific to cue, strong bar-grip-reflex |
| 5 | Completely lethargy, body position is not self-determined, decreased bar-grip-reflex |

## Sample size estimation, experimental repetition, group allocation

Sample size estimation for time-of-da- dependent endotoxic shock experiments has been performed by running Monte Carlo simulations. Parameters of this simulations were: Amplitude of time-of-day-dependent mortality (ranging from 0% - no amplitude, to 40% - peak mortality 90%, trough mortality 10%), number of mice per time point (ranging from 1 to 32), number of time points (ranging from 3 to 24). Parameter settings in have been simulated in discrete steps with 1000 simulations per combination. Final sample size was chosen from parameter combination, which allowed us to detect a 20% amplitude of mortality (determined by significant p-value as analyzed by Chronolyse program) with a chance of 50%. Unless otherwise stated in figure legends, animal experiments have been performed once, considering each mouse as a biological replicate. Mice have been randomly assigned to experimental groups (i.e. time points of LPS administration) and subsequently masked for scoring by an independent observer.

## Peripheral blood cytokine concentrations in endotoxic shock

To determine the blood cytokine levels in the endotoxic shock model, mice were injected with corresponding $LD_{50}$ of LPS at six 4 hr spaced time points (n=14). Two hr post LPS injection mice were terminally bled by cardiac puncture using a 23G x 1' cannula (Henke-Sass Wolf). Syringes (1 ml, Braun) were coated with heparin (Ratiopharm) to avoid blood coagulation. After isolation blood was kept at 4°C for subsequent plasma preparation. To this end, blood was centrifuged for 15 min, 370 g at 4°C. Plasma was isolated, aliquoted and frozen at −80°C for further analysis.

## Isolation of peritoneal macrophages (PM)

Mice were sacrificed by cervical dislocation. Peritoneal cavity cells (PEC) were isolated by peritoneal lavage with ice cold PBS. Lavage fluid visibly containing red blood cells was dismissed. For RNA measurements or bioluminescence recordings, peritoneal macrophages were further purified by MACS sorting (Miltenyi) according to manufacturer's protocol using mouse/human CD11b Microbeads and LS columns. Eluates containing positively sorted macrophages were analyzed for sorting efficiency by FACS. All steps were performed at 4°C.

## Isolation of bone marrow cells

Mice were sacrificed by cervical dislocation. One tibia and femur were excised per mouse. Femur and tibia were flushed with supplemented RPMI 1640 and erythrocytes were lysed using GEYS solution (2 min at 4°C). After erythrocyte lysis, cells were suspended in supplemented RPMI 1640 medium and filtered through a 30 μM filter (Miltenyi). Cell numbers were determined using a Neubauer chamber. All steps were performed at 4°C. All centrifugation steps were performed at 4°C, 300 g, 7 min.

## Isolation of spleen cells

Mice were sacrificed by cervical dislocation. Spleen was removed and a single-cell suspension was obtained using gentleMACS, Miltenyi (program: m-spleen-01) with C-tubes (Miltenyi) in PBS. Next, single-cell solution was filtered using 100 μM cell strainers (Thermo Fischer). GEYS solution was used for erythrocyte lysis for 2 min at 4°C. After erythrocyte lysis cells were suspended in supplemented RPMI 1640 and filtered through a 30 μM filter (Miltenyi). Cell number was determined using a Neubauer chamber. All steps were performed at 4°C. All centrifugation steps were performed at 4°C, 300 g, 7 min.

## Bioluminescence recordings

PER::LUC protein bioluminescence recordings were used to characterize circadian clock function in peritoneal macrophages, SCN and lung tissue. Mice were sacrificed by cervical dislocation. Brains and lungs were isolated and transferred to chilled Hank's buffered saline solution, pH 7.2. (HBSS). For tissue culture, 300 μm coronal sections of the brain and 500 μm sections of the lung were obtained using a tissue chopper. The lung and SCN slices were cultured individually on a Millicell membrane (Millipore) in a Petri-dish in supplemented DMEM containing 1 μM luciferin (Promega). PECs were isolated by peritoneal lavage and immediately CD11b-MACS sorted. The CD11b-sorted peritoneal macrophages were cultured in Petri dishes in supplemented DMEM containing 1 μM

luciferin (Promega). For bioluminescence recording, tissues/primary cell cultures were placed in light-tight boxes (Technische Werkstaetten Charite, Berlin, Germany) equipped with photo-multiplier tubes (Hamamatsu, Japan) at standard cell culture conditions. Bioluminescence was recorded in 5 min bins. On fifth day in culture, PMs were treated with 1 µM dexamethason for 1 hr, followed by a medium change to supplemented DMEM containing 1 µM luciferin (Promega). Data were further processed and analyzed using Chronostar 3.0 *Maier et al., 2021*.

## Generation of whole cell protein lysates

Liver and peritoneal cavity cells were homogenized in ice cold RIPA buffer containing 1x protease-inhibitor-cocktail (Sigma Aldrich) and incubated on ice for 30 min. Homogenized cells were then centrifuged at maximum speed for 30 min at 4°C to pellet the insoluble cell debris. The supernatant fraction was then removed and used for cellular protein analysis or frozen at −80°C. Protein concentrations were determined using standard BCA assay.

## Immunoblotting

Samples were denatured for SDS-PAGE in NuPAGE SDS Sample Buffer (4x) (Invitrogen) containing 0.8% 2-$\beta$-mercaptoethanol (Sigma) and boiled for 5–10 min at 95°C. SDS-PAGE using 4–12% Bis-Tris gels (Thermo Scientific) at 200V for 60 min in NuPAGE MES SDS Running Buffer. Proteins were transferred to a nitrocellulose membrane (0.45 µm) using a tank transfer system (wet transfer). NuPAGE transfer buffer, containing 20% Methanol, was cooled with an ice block to prevent overheating during the transfer. The transfer was run for 120 min at 90V. Following the transfer, the membrane was blocked in TBS-T with 5% non-fat, dry milk for 1–2 hr at RT. After a washing step in TBS-T (3 x 10 min), the membrane was placed in the primary antibody solution (TBS-T with 5% non-fat, dry milk) and gently shaken overnight at 4°C. The membrane was then washed in TBS-T (3 x 10 min) and incubated with the HRP-conjugated secondary antibody (Santa Cruz Biotechnologies) in TBST-T for 2 hr at RT. After another washing step in TBS-T (3 x 10 min), a chemiluminescence reaction was performed with Super SignalWest Pico substrate (Pierce). The protein bands were visualized using the ChemoCam detection system (Intas). The following primary antibodies were used: murine CLOCK - rabbit anti-mCLock (Bethyl Laboratories, A302-618A), murine BMAL1 - rabbit anti-mBMAL1 (kind gift from Micheal Brunner), murine ACTINB - mouse anti mBactin (Sigma, A5441). Secondary antibodies used: goat anti-mIgG-HRP (SantaCrz Biotechnology, sc-2005), donkey anti-rbIgG-HRP (SantaCrz Biotechnology, sc-2005).See also key resource table for additional information.

## Single- and multiplex immunoassays

Singleplex assay: murine IL-6 plasma concentration was determined by ELISA according to manufacturer's protocol (Ebioscience) in a 96-well format (Corning). Plasma samples were diluted 1:200 in supplied assay buffer. Absorption was measured at 470 nm. Reference wavelength was measured at 560 nm by Infinite F200Pro plate reader (Tecan). Multiplex assay: 13 cytokines (CCL2 / MCP-1, CCL3 / MIP-1α, CCL4 / MIP-1β, CCL7 / MCP-3, CXCL5, IL-1α, IL-1β, IL-10, IL-12p40, IL-18, Eotaxin, Rantes / CCL5, TNFα) were assessed using the ProCartaPlex, Mix and Match, Mouse 13-Plex (Affymetrix, eBioscience). ProCartaPlex was performed as described in manufacturer's protocol in a 96 well format (eBioscience). All washing steps were performed using a hand held magnetic washer (eBioscience). Data were acquired using a MagPix (Luminex) detection device. Data evaluation was performed using ProcartaPlex Analyst v.1.0 (eBiosciences).

## Isolation and quantification of RNA

Total RNA was isolated using the PureLink RNA Mini Kit (Ambion) according to the manufacturer's manual. In addition, an on-column DNA digestion was performed using PureLink DNase Set (Life Technologies). RNA was quantified by measuring the absorption at 260 nm with NanoDrop 2000C (Thermo Scientific).

## Quantitative real-time PCR

Total RNA was reverse-transcribed to cDNA using random hexamers to prime reverse transcriptase reaction. cDNA was diluted 1:10 in H$_2$O for use in qRT-PCR. qRT-PCR was performed using a two-step protocol with the following primer-sets: primer-sets for mCry1, mCry2, mDpb, mNr1d1,

mNpas2, mPer1, mPer2 were purchased from Qiagen (QT00117012, QT00168868, QT00103089, QT00164556, QT00108647, QT00113337, QT00198366, respectively). mGapdh, fw: ACGGGAAGC TCACTGGCATGGCCTT, rv: CATGAGGTCCACCACCCTGTTGCTG; mBmal1 primers were designed to characterize myBmal-KO mice. Forward primer (fw: GGACACAGACAAAGATGACCC) binds upstream of exon eight and the reverse primer (rv: TTTTGTCCCGACGCCTCTTT) within exon 8 of *Bmal1*. Thus after successful Cre recombination, exon eight is deleted and no PCR product is detectable. *Clock* primers were designed to characterize myClock-KO mice. Primers bind in exon 5 (fw: A TTGGTGGAAGAAGATGACAAGGA) and in exon 6 (rv: TACCAGGAAGCATAGACCCC) of clock. As exon 5 and 6 are flanked by loxP sites, after successful Cre recombination no PCR product is amplified.

## Flow cytometry (FACS)

Two panels of antibodies were established to target a broad range of immune cells in various sites of the organism. Before each experiment, antibody mix of both panels were prepared and kept on 4°C for labeling of all samples of the respective experiment in order to minimize intra-experimental variability. All antibody-mixes were prepared in FACS buffer containing 1:50 FcR blocking reagent. FACS staining of samples: 100 µL of the cell suspensions were transferred into a 96-well plate and spun down for 7 min, 300 g at 4°C. Supernatant was carefully discarded and pellet re-suspended in 50 µL of master-mix containing one of two antibody panels. Cells were incubated for at least 30 min at 4°C in darkness. Subsequently 200 µL FACS buffer was added and cells were centrifuged for 7 min, 300 g at 4°C. Cells were washed twice using 200 µL FACS buffer each before fixation in 200 µL 4% PFA for 30 min at RT. Finally, cells were spun down and re-suspended in 200 µL FACS buffer and stored at 4°C for up to a week before FACS data acquisition in a FACS CantoII (BD Biosciences).

## Statistical data analysis

Statistical analysis was performed in GraphPad Prism 8 and R. Normality was tested using Shapiro-Wilk normality test. When data were normally distributed and two parameters were compared, one-way ANOVA with Dunnett's multiple comparison as a post hoc test was applied. When comparing more than two groups and two parameters a two-way ANOVA with Bonferonni's post hoc test was applied. Two sample comparison was performed with two-sided students t-test for normally distributed data and Mann-Whitney U test for non-normally distributed data. Time-of-day dependent mortality experiments: Mortality data were transformed to probability of death (between 0 and 1) in order to compute sine fit using alogistic regression. The confidence interval was derived from sine fit estimation. After statistical analysis, the mortality data was transformed back to the initial percentages. Cross-correlation analysis of morality and cytokine data: To correlate mortality rates and cytokine levels across all animal models, a permutation of sine fit of the mortality data and plasma cytokine data was used. This bootstrapping (randomization) procedure gives rise to the empirical distribution of correlations. The p-value is the fraction of randomizations that gave a correlation with the opposite denominator, meaning that a p=0.05 means that only 5% of correlations crossed zero correlation threshold. For linear regression analysis of the summary of cytokine and mortality data, the estimated mortality rate determined by sine fit of mortality data was correlated to mean value of plasma cytokine concentration using Spearman's rank correlation. Bioluminescence recordings were analysed using in-house written software ChronoStar 3.0 (*Maier et al., 2021*). In brief, raw bioluminescence counts were transformed to log-space and trends removed by subtracting the 24 hr running average. Circadian rhythm parameters were estimated by fitting a damped sine wave to these data. Finally, data were reversely transformed into linear space. Circadian rhythmicity of cytokine time-series was tested using our in-house written software ChronoLyse. In brief, a 24 hr sine wave was fitted the beforehand log-transformed data and parameters of fit were used to estimate amplitude, phase and mean levels. Rhythmicity was tested by testing against a flat line using F-test.

## Acknowledgements

This work was supported by the Deutsche Forschungsgemeinschaft (DFG, Grants AN 1553-2/1, MA 5108/1–1, HE2168/11-1, SPP 2041).

## Additional information

### Funding

| Funder | Grant reference number | Author |
| --- | --- | --- |
| Deutsche Forschungsgemeinschaft | MA 5108/1-1 | Bert Maier |
| Deutsche Forschungsgemeinschaft | AN 1553-2/1 | Bharath Ananthasubramaniam |

The funders had no role in study design, data collection and interpretation, or the decision to submit the work for publication.

### Author contributions

Veronika Lang, Formal analysis, Validation, Investigation; Sebastian Ferencik, Investigation; Bharath Ananthasubramaniam, Formal analysis; Achim Kramer, Conceptualization, Resources, Supervision, Writing - review and editing; Bert Maier, Conceptualization, Data curation, Formal analysis, Supervision, Funding acquisition, Investigation, Visualization, Methodology, Writing - original draft, Project administration, Writing - review and editing

### Author ORCIDs

Bharath Ananthasubramaniam (iD) http://orcid.org/0000-0003-4467-1546
Bert Maier (iD) https://orcid.org/0000-0002-5254-008X

### Ethics

Animal experimentation: All procedures were authorized by and performed in strict accordance with the guidelines and regulations of the German animal protection law (Deutsches Tierschutzgesetz). The protocols were approved by the ethics comittee of the Landesamt für Gesundheit und Soziales (LaGeSo, Permit Number G 0161/12 and G 0211/14).

### Decision letter and Author response

Decision letter https://doi.org/10.7554/eLife.62469.sa1
Author response https://doi.org/10.7554/eLife.62469.sa2

## Additional files

### Supplementary files

- Source data 1. Folder containing all article related GraphPad figure files.

- Transparent reporting form

### Data availability

All data generated or analysed during this study are included in the manuscript and supporting files.

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
