## [Decision Letter]

**Acceptance summary:**

This manuscript has novelty in its approach. The authors use an animal model to abolish the circadian rhythm in mice to study the impact on susceptibility to challenge with LPS. The experimental approach they use involves both wild-type mice subject to sudden stop of the light-dark (LD) cycle and mice knocked-out for the Clock system (KO).

**Decision letter after peer review:**

Thank you for submitting your article "Susceptibility rhythm to bacterial endotoxin in myeloid clock-knockout mice" for consideration by *eLife*. Your article has been reviewed by 3 peer reviewers, including Evangelos J Giamarellos-Bourboulis as Reviewing Editor and Reviewer #1, and the evaluation has been overseen by Satyajit Rath as the Senior Editor.

The reviewers have discussed the reviews with one another and the Reviewing Editor has drafted this decision to help you prepare a revised submission.

As the editors have judged that your manuscript is of interest, but as described below that additional experiments are required before it is published, we would like to draw your attention to changes in our revision policy that we have made in response to COVID-19 (https://elifesciences.org/articles/57162). First, because many researchers have temporarily lost access to the labs, we will give authors as much time as they need to submit revised manuscripts. We are also offering, if you choose, to post the manuscript to bioRxiv (if it is not already there) along with this decision letter and a formal designation that the manuscript is "in revision at eLife". Please let us know if you would like to pursue this option. (If your work is more suitable for medRxiv, you will need to post the preprint yourself, as the mechanisms for us to do so are still in development.)

Summary:

This manuscript has novelty in its approach. The authors investigate and document the role of myeloid-endogenous circadian cycling on the host response to and progression of endotoxemia in the mouse LPS-model. As a principal finding, Lang et al., report how disruption of the cell-intrinsic myeloid circadian clock by myeloid-specific knockdown of either CLOCK or BMAL1 does not prevent circadian patterns of morbidity and mortality in endotoxemic mice. As a consequence of these and other findings from endotoxemia experiments in mice kept in the dark or the observation of circadian cytokine production in CLOCK KO animals, the authors conclude that myeloid responses critical to endotoxemia are not governed by their local cell-intrinsic clock. Moreover they conclude that the source of circadian timing and pace giving that is critical for the host response to endotoxemia must lie outside the myeloid compartment. Finally, the authors also report a general (non-circadian) reduced susceptibility of mice devoid of myeloid CLOCK or BMAL1, which they take as a proof that myeloid circadian cycling is important in the host response to endotoxemia, yet does not dictate the circadian pattern in mortality and cytokine responses.

The paper is well conceived, experiments are very elegant and well carried out, statistics and ethics statements are appropriate. The conclusions of this study, as summarized above, are important and will be of much interest to readers from the circadian field and beyond, also to sepsis and inflammation researchers. There is one major flaw in the argumentative line of this story, as the study relies on the assumption that the systemic cytokine response provided by myeloid cells is paramount and central to the course and intensity of endotoxemia. While this is assumed, rigorous proof of this connection and its causality is still lacking (most evidence is of correlative nature). This is important, since there is an increasing body of more recent experimental evidence that argues against a prominent role of myeloid cells in the cytokine storm. While the intriguing findings are interesting, the study remains mainly descriptive and frustratingly without mechanistic explanation. At least, the following revisions are strongly recommended.

Essential revisions:

1. The investigators show that mice shift from LD to DD become more lethal to LPS. If this is due to abolishment of the circadian rhythm, similar lethality should appear with the challenge of the KO mice. The opposite was found. Please explain.

2. LPS is acting through TLR4 binding. Can the author provide evidence that TLR4 expression is down-regulated in transition from LD to DD? Does the same apply for the expression of SOCS3?

3. As mentioned, a central weakness of this paper is that it assumes systemic cytokine levels as produced by myeloid cells are center stage in endotoxemic shock (e.g. see line 164). However, recent evidence has shown that over 90% of most of systemically released cytokines in sepsis can be produced by non-myeloid cells (as shown e.g. by use of humanized mice), which allows discrimination between (human) cytokines produced by blood cells from (murine) cytokines produced by parenchyma (see e.g. PMID: 31297113). (Interestingly, there is a one major exception to that rule, and that is TNFa). Considering this, it is not surprising that circadian cytokine levels do not change in myeloid CLOCK/BMAl1 KO mice. Also, assuming that myeloid-produced cytokines are not critical drivers, the same applies to the observation that circadian mortality pattern is preserved in those mice. The authors should more critically discuss this alternative explanation. In fact, this line of arguing would be in line with the concept that the source for the circadian susceptibility /mortality in endotoxemia resides in a non-myeloid cell compartment, which is essentially the major finding of this manuscript.

4. Intro (lines 51-54): the authors describe one scenario as the mechanism of sepsis-associated organ failure. This appears too one-sided and absolute, since many more hypothesis and models exist. It would be good to mention that and/or tone down the wording.

5. Very analogous to Light/Darkness cycles, ambient temperature has been shown to have a strong impact on mortality from endotoxemia (e.g. PMID: 31016449). Did the authors keep their animals in thermostated ambient conditions? Please describe and discuss in the text.

6. Figure 2C; The large difference in mortality in the control lys-MCre line looks somewhat worrying. Could this be a consequence of well-known Cre off-target activities? Did the authors check this by e.g. sequencing myeloid cells of or using control mouse strains?

7. Line 320: Bmal1flox/flox (Bmal-flox) [48] or Clockflox/flox (Clock-flox) [38] were bred with LysM-Cre to target Bmal1. Please show a prototypical genotyping result, perhaps as supplemental figure.

8. Line 365: the authors state that mice that did not show signs of disease were sorted out. What proportion of mice (%) did not react to LPS? It would be useful to state this number in the methods section.

9. It is not fully clear if male or female or both were used for the principal experiments, please specify. If female were used, please describe how estrous cycle was taken into account.

10. Please revise the result section and the legends (for example legends of Figures 3 and 5) to explicitly mention whether experiments with conditional knockouts were performed with LD or DD mice.

11. Specific issues:

a. Line 15 and 80. Saying that DD mice show a "three-fold increased susceptibility to LPS" is true for very specific conditions only, and should not be used as a general statement.

b. Line 99-. Please be more precise in describing cytokine levels (for example, in LD, TNF peaks at ZT10, IL-18 at ZT14 or ZT22 but not ZT18, and IL-10 but not IL-12 peaks at ZT14).

c. Line 105-106. Referring to Figure 1E, it is not straightforward for the reader to understand what is meant by "free-running and entrained" conditions.

d. Figure 5 and Sup Figure 5. They are huge differences in leukocytes counts between LysM-Cre+/+ and WT mice. Without being exhaustive, LysM-Cre+/+ display much more macrophages in bone marrow, spleen and lymph nodes, DCs in lymph nodes, NK cells in spleen and lymph nodes at both CT8 and CT20. This is very puzzling and questions about the pertinence of these "control" mice. Additionally, one might expect from these observations that LysM-Cre+/+ mice are more sensitive to endotoxemia, which is not the case (point 5).

e. Line 257. The effect of IL-18 is not totally surprising, since both detrimental and protective effects of the cytokine have been reported in the literature. This could be briefly mentioned.

f. Sup Figure 5A. The gating strategy has to be shown for each organ, separately.

g. Sup Figure 5D. The peritoneal cavity contains not only different macrophage populations with different inflammatory properties, but also different B cell populations including anti-inflammatory B-1a cells (plus NK cells, DCs…). Considering that LPS is injected i.p., more thorough analyses of the peritoneal cavity should performed to properly interpret results of cytokine and mortality.

h. It is not clear whether endotoxemia was addressed with BMAL1 and CLOCK myeloid conditional knockout mice kept LD. Since time-of-day dependent differences in mortality were much less in DD mice (line 74), we probably expect only marginal differences in DD mice.

---

## [Author Response]

Essential revisions:1. The investigators show that mice shift from LD to DD become more lethal to LPS. If this is due to abolishment of the circadian rhythm, similar lethality should appear with the challenge of the KO mice. The opposite was found. Please explain.

We thank the reviewer for pointing to this misleading passage. The reviewer correctly states, that mice switched from LD to DD show higher mortality to LPS (Figure 1 A and B). However, neither control mice nor myeloid Clock knockout mice have abolished circadian rhythms, as demonstrated in Figure 3—figure supplement 1 A.

In addition, the observation that mice are more susceptible to LPS after transition from LD to DD was also made in the myBmal1-KO strain (Figure 2 D).

However, the underlying reason for this increase in susceptibility is not known and needs further investigation as we discuss in lines 320ff.

We have added and modified the text in the manuscript in line 135ff and line 321 to improve clarity.

2. LPS is acting through TLR4 binding. Can the author provide evidence that TLR4 expression is down-regulated in transition from LD to DD? Does the same apply for the expression of SOCS3?

We thank the reviewer for this interesting question. Since both, TLR4 and SOCS3 are mainly regulated on transcriptional and post-transcriptional level (Yan, 2006; Yoshimura et al., 2007) we expected to see potentially regulated expression on mRNA level. We thus performed an experiment investigating peritoneal cavity cells from mice housed either in LD or after transition from LD to DD at two time points (ZT/CT8 and ZT/CT20). We chose these time points since we expected to observe highest levels of regulation of TLR4/SOCS3 then, in case these genes were regulated in a time of day dependent manner.

While in both light treatment groups *Nr1d1* showed the expected time of day dependent differences in expression (p-values: a = 0.09 , b = 0.04 ), *Tlr4* expression levels remained unaltered between time points (not tested) and light treatment (p-values: c = 0.63 , d = 0.59 ). For Socs3 a trend towards differences between time points (p-values: g = 0.20 , h = 0.33 ) but not between light treatment groups (p-values: e = 0.62, f = 0.91 ) could be observed. All error bars represent SEM, n=3 (ZT8) or n=5 (other conditions). Thus, these data provide no evidence that Tlr4 of Socs3 are differentially regulated upon transition from LD to DD and therefore cannot explain the higher lethality of LPS in mice after transfer to DD.

3. As mentioned, a central weakness of this paper is that it assumes systemic cytokine levels as produced by myeloid cells are center stage in endotoxemic shock (e.g. see line 164).

We thank the reviewer for this comment, which uncovers that we need a clearer description of the underlying model and hypothesis of this paper. We added and changed a passage in the introduction for a more detailed description of the mechanisms leading to endotoxic shock (lines 55ff).

However, recent evidence has shown that over 90% of most of systemically released cytokines in sepsis can be produced by non-myeloid cells (as shown e.g. by use of humanized mice), which allows discrimination between (human) cytokines produced by blood cells from (murine) cytokines produced by parenchyma (see e.g. PMID: 31297113). (Interestingly, there is a one major exception to that rule, and that is TNFa). Considering this, it is not surprising that circadian cytokine levels do not change in myeloid CLOCK/BMAl1 KO mice. Also, assuming that myeloid-produced cytokines are not critical drivers, the same applies to the observation that circadian mortality pattern is preserved in those mice.

We agree with the reviewer, that myeloid cells are neither the only source, nor the source of the majority of cytokines found in plasma upon LPS challenge. However, there is broad consensus in the field that cells of the myeloid lineage are among the first sensors of LPS and therefore may be critical in determining the avalanche-like cytokine reaction including other cell types such as endothelial cells.

The reviewer mentions an interesting work (Skirecki et al., 2019), which investigates the early outcome-dependent inflammatory response in a humanized mouse model after cecal ligation and puncture induced sepsis. The findings of this study provide first numbers for the contribution of different cellular sources of cytokines in plasma before and after sepsis induction. However, the humanized mouse model does not allow to discriminate between first and second order cytokine responses and therefore has limited implications on current models, which assigns myeloid lineage cells and the complement system a central role in LPS sensing (Hotchkiss et al., 2016; Laudanski, 2021).

We also consider it as a strength of our paper that we challenged a circulating hypothesis based on a circadian mortality pattern upon LPS challenge in mice. While the results of our work cannot uncover the origin of this pattern, it clearly can rule out a pivotal role for myeloid clocks in its generation.

The authors should more critically discuss this alternative explanation. In fact, this line of arguing would be in line with the concept that the source for the circadian susceptibility /mortality in endotoxemia resides in a non-myeloid cell compartment, which is essentially the major finding of this manuscript.

We would like to thank the reviewer for his or her constructive critique and as suggested, we extended the discussion on alternative hypotheses on the contribution of early, fate decisive factors in circadian mortality by endotoxic shock (lines 288ff).

4. Intro (lines 51-54): the authors describe one scenario as the mechanism of sepsis-associated organ failure. This appears too one-sided and absolute, since many more hypothesis and models exist. It would be good to mention that and/or tone down the wording.

We thank the reviewer for this comment and extended the section about the pathophysiology of septic shock to picture a more differentiated model of the mechanisms leading to death (see also point 3 (lines 55ff)).

5. Very analogous to Light/Darkness cycles, ambient temperature has been shown to have a strong impact on mortality from endotoxemia (e.g. PMID: 31016449). Did the authors keep their animals in thermostated ambient conditions? Please describe and discuss in the text.

We thank the reviewer for this comment. Animals have been kept in a thermostated, light and humidity controlled mouse facility. We added the specific set parameters in the methods (Clausen et al., 1999) (lines 336ff).

6. Figure 2C; The large difference in mortality in the control lys-MCre line looks somewhat worrying. Could this be a consequence of well-known Cre off-target activities? Did the authors check this by e.g. sequencing myeloid cells of or using control mouse strains?

We agree with the reviewer, that the large differences in mean mortality in the WT and lysM-cre lines are unexpected and partially weaken the effect size on reduced mean mortality in the myeloid clock knockout animals. However, we would like to emphasize that there is no statistical significant difference despite a rather large number of tested animals between the WT and the lysM-cre control mice.

Cre-lines are known to display a variety of hardly predictable side effects, namely cre-toxicity, target gene efficiency (Abram et al., 2015) and off-target gene activity (infidelity, Schmidt-Supprian and Rajewsky, 2007). In case of the LysM-cre mice, the two latter have been investigated by Clausen (Clausen et al., 1999). The authors conclude that the cre-activity is highly specific for myeloid cells and has an efficiency of about 80 % for monocytes / macrophages and more than 90 % for granulocytes. Similar values have been found by Abram et al. Cre-toxicity as the enzymatic attacking of so-called genomic pseudo-loxP sites has not been described so far for the LysM-cre mice. However, we cannot rule out such a possibility and there exist only few measures to control for such effects (Schmidt-Supprian and Rajewsky, 2007): 1. high level Cre expression for long time spans should be avoided – and this seems to be the case in the LysM-cre mice according to Claussen et al. 2. A LysM-cre control should be included in the analysis of conditional knock-out mice and this is exactly what we did in figure 2C. 3. The utilization of two different floxed mouse lines targeting the same transcription factor complex (Clock/Bmal1) further controls for circadian clock specific effects in our conditional knock-out experiments.

7. Line 320: Bmal1flox/flox (Bmal-flox) [48] or Clockflox/flox (Clock-flox) [38] were bred with LysM-Cre to target Bmal1. Please show a prototypical genotyping result, perhaps as supplemental figure.

We added Figure 3—figure supplement 1 C to show prototypical genotyping results for Clock-flox and Figure 2—figure supplement 1 B for Bmal1-flox mice.

8. Line 365: the authors state that mice that did not show signs of disease were sorted out. What proportion of mice (%) did not react to LPS? It would be useful to state this number in the methods section.

Five of 632 mice (0.8%) did not show signs of disease. We added this information in the methods section (lines 404-405).

9. It is not fully clear if male or female or both were used for the principal experiments, please specify. If female were used, please describe how estrous cycle was taken into account.

Only male mice have been used for all the experiments in this paper. We apologize that we missed to provide this important information at a more prominent place. We now added this information also in the methods in section 'Endotoxic shock experiments' (line 376).

10. Please revise the result section and the legends (for example legends of Figures 3 and 5) to explicitly mention whether experiments with conditional knockouts were performed with LD or DD mice.

We thank the reviewer for pointing to these weaknesses in our experimental descriptions. We added this information at various places in the results and legends text (i.e. lines 130, 159, 167, 183, 216 and legends of Figure 3 C, F, G and Figure 5 C-E) and think that the manuscript is more comprehensible by that information.

11. Specific issues:a. Line 15 and 80. Saying that DD mice show a "three-fold increased susceptibility to LPS" is true for very specific conditions only, and should not be used as a general statement.

We now see that the phrase "susceptibility to LPS-induced mortality" can be misleading. We therefore rephrased it to "susceptibility to LPS" (line 87), which should allow to not only compare the endpoints (i.e. mortality levels) but rather the sensitivity of the system at various levels of LPS dosing. In addition, we specified the interpretation of our data as being valid for the investigated conditions (lines 86ff).

b. Line 99-. Please be more precise in describing cytokine levels (for example, in LD, TNF peaks at ZT10, IL-18 at ZT14 or ZT22 but not ZT18, and IL-10 but not IL-12 peaks at ZT14).

We thank the reviewer for pointing to this ambiguous description of our data. When we thought about how to describe these data, we felt that the phases (peak times, Figure 1F) obtained by sinusoidal fitting might be better suited to provide general patterns than the actual values (Figure 1 E). Thus, instead of describing the true measured peak-times, we used the values, which have been determined by interpolation based on a sinusoidal -fit. We still believe that our way to describe these cytokine data is scientifically sound. To avoid reader confusion we have modified the relevant text and added some words to clarify the context (lines 108 + 109).

c. Line 105-106. Referring to Figure 1E, it is not straightforward for the reader to understand what is meant by "free-running and entrained" conditions.

We thank the reviewer for pointing to this wording, which we apologize that it is written in "chronobiology" jargon. Since we clearly want to appeal to a broader readership with this work, we have added the information about the applied light conditions (lines 114 + 115).

d. Figure 5 and Sup Figure 5. They are huge differences in leukocytes counts between LysM-Cre+/+ and WT mice. Without being exhaustive, LysM-Cre+/+ display much more macrophages in bone marrow, spleen and lymph nodes, DCs in lymph nodes, NK cells in spleen and lymph nodes at both CT8 and CT20. This is very puzzling and questions about the pertinence of these "control" mice. Additionally, one might expect from these observations that LysM-Cre+/+ mice are more sensitive to endotoxemia, which is not the case (point 5).

We fully agree with the reviewer, that the differences between the two control mouse strains in the two time-point experiments presented in figure 5 are huge, puzzling and contradicted our expectations. In part, these differences are diminished after we thoroughly reanalysed the FACS data as a consequence of errors in our previous gating settings. Still, large differences between LysM-Cre and WT mice of leukocyte subtype counts in various tissues remain (e.g. macrophages in spleen and lymphnodes, neutrophils in lymphnodes, CD4 T-cells in spleen).

Nevertheless, we consider the LysM-Cre mice to the most appropriate controls for the myeloid clock knock-out mouse strains (see also response to point 5). In our opinion, there is one point, which could be improved in a similar experiment – but we decided to not implement it for logistical and ethical reasons: If we had bred the knockout strains of the myeloid clock on a heterozygous genetic background for both floxed and cre alleles, it would have been possible for us to use littermate controls and thus also control genetic drift and the exact breeding conditions.

We would also like to mention that for one specific experimental aspect, namely the time-of-day dependent traffic of myeloid cell types, relative numbers between circadian time points and not between mouse strains are important.

e. Line 257. The effect of IL-18 is not totally surprising, since both detrimental and protective effects of the cytokine have been reported in the literature. This could be briefly mentioned.

We agree with the reviewer, that the protective role of IL-18 has been described previously (for a review see Dinarello et al., 2013). However, to our knowledge, these reports are based ona colitis model in mice, and we could not find reports about a protective role for IL-18 in sepsis.

We now mention this in the discussion in lines 271+272

f. Sup Figure 5A. The gating strategy has to be shown for each organ, separately.

We thank the reviewer for this comment and added the gating strategies for every organ separately in the new Figure 5—figure supplements 1-6.

g. Sup Figure 5D. The peritoneal cavity contains not only different macrophage populations with different inflammatory properties, but also different B cell populations including anti-inflammatory B-1a cells (plus NK cells, DCs…). Considering that LPS is injected i.p., more thorough analyses of the peritoneal cavity should performed to properly interpret results of cytokine and mortality.

We thank the reviewer for this suggestion and agree with his or her assessment about the cellular composition of peritoneal cavity cells. Unfortunately, our lymphoid antibody panels used to analyze B, T, and NK cell populations did not allow finer differentiation for sub-B cell populations because of our preference for subpopulations of myeloid origin. We hope that the results shown in Figure 5—figure supplement 9 and 10 are nevertheless a sufficient representation of the major cell types in the peritoneal cavity to allow a thorough interpretation of our results.

h. It is not clear whether endotoxemia was addressed with BMAL1 and CLOCK myeloid conditional knockout mice kept LD. Since time-of-day dependent differences in mortality were much less in DD mice (line 74), we probably expect only marginal differences in DD mice.

The point raised by the reviewer is related to point 8, as we failed to explicitly describe the light conditions for each experiment at the required locations. The endotoxemia experiments using myBmal-KO (Figure 2 A ) or myClock-KO mice (Figure 3 F) were performed in constant darkness (DD).

One of the main conclusions of our work presented in this manuscript is, that time-of-day dependent differences are equally pronounced in mice housed in LD and in DD. The reduction of these differences observed in Figure 1 A is due to a "ceiling" effect i.e. increased susceptibility in mice housed in DD.

We have added the information on the light conditions for each experiment unless this has been done explicitly before (see also comment on point 9).